# Investigating the associations between intimate partner violence and nutritional status of women in Zimbabwe

**Jeanette Iman'ishimwe Mukamana**[1], **Pamela Machakanja**[1], **Hajo Zeeb**[2,3], **Sanni Yaya** [4], **Nicholas Kofi Adjei** [2,3,5]*

**1** Institute of Peace, Leadership and Governance, Africa University, Mutare, Zimbabwe, **2** Leibniz Institute for Prevention Research and Epidemiology – BIPS, Bremen, Germany, **3** Health Sciences Bremen, University of Bremen, Bremen, Germany, **4** School of International Development and Global Studies, Faculty of Social Sciences, University of Ottawa, Ottawa, Canada, **5** Department of Public Health, Policy and Systems, University of Liverpool, Liverpool, United Kingdom

* n.adjei@liverpool.ac.uk

**Data Availability Statement:** The data used for this study were obtained from the Demographic and Health Survey (DHS), https://dhsprogram.com/. Access to this data is free of charge; however, any

## Abstract

### Background

Intimate partner violence (IPV) against women and poor nutritional status are growing health problems in low and middle-income countries (LMICs). Moreover, violence against women has been shown to be associated with poor nutrition. This study investigated the relationship between IPV and nutritional status (i.e., underweight, overweight, and obesity) among women of reproductive age (15–49 years) in Zimbabwe.

### Methods

Pooled data from the 2005/2006, 2010/2011, and 2015 Zimbabwe Demographic Health Surveys (ZDHS) on 13,008 married/cohabiting women were analysed. Multinomial logistic regression models were used to examine the associations between the various forms of IPV and the nutritional status of women. We further estimated the prevalence of BMI $\geq$ 25.0 kg/m$^2$ (overweight and obesity) by intimate partner violence type.

### Results

The mean BMI of women was 24.3 kg/m$^2$, more than one-fifth (24%) were overweight and about 12% were obese. Forty-three percent (43%) of women reported to have ever experienced at least one form of intimate partner violence. More than one-third (35%) of women who reported to have ever experienced at least one form of intimate partner violence had a BMI $\geq$ 25.0 kg/m$^2$ (p< 0.01). Relative to normal weight, women who had ever experienced at least one form of IPV (i.e., physical, emotional, or sexual) were more likely to be obese (aOR = 2.59; 95% CI = 1.05–6.39). Women's exposure to any form of intimate partner violence was not significantly associated with the likelihood of being underweight or overweight relative to normal weight.

researcher wishing to use the data must register to have access.

**Funding:** The authors received no specific funding for this work.

**Competing interests:** The authors have declared that no competing interests exist.

## Conclusions

The study findings show that women of reproductive age in Zimbabwe are at high risk of both IPV and excess weight. Moreover, we found a positive relationship between exposure to at least one form IPV and obesity. Public health interventions that target the well-being, empowerment and development of women are needed to address the complex issue of IPV and adverse health outcomes, including obesity.

## Introduction

Intimate partner violence (IPV) is a form of gender-based violence [1], mostly perpetrated against women [2,3]. This behaviour is assaultive and coercive [4,5], and it comes in the form of emotional, sexual, or physical abuse [6–9]. The various forms of abuse may co-exist [10]; for instance, physical abuse or violence is mostly accompanied by sexual violence, and the latter may also come along with emotional violence [10,11]. IPV is increasingly recognized as a relevant social and health problem by relevant institutions and organizations worldwide [12–14], due to its adverse impacts on victims [10,15], and society as a whole [7,16–19].

The prevalence of IPV is high in developing countries [1]. However, there is evidence of cross-country variations [20,21], where Zimbabwe has been found to be one of the countries in sub-Saharan Africa with the highest prevalence of IPV [21,22]. It is estimated that approximately 35% of women had experienced physical violence from the age of 15 and 14% had experienced sexual violence [23]. In a recent study in Zimbabwe, Mukamana and colleagues found a substantial rise in the prevalence of IPV from 40.9% in 2010 to 43.1% in 2015 [1].

Violence against women as a health problem [16,17] has been shown to be one of the leading causes of both medical diagnosed and non-medical explainable physical, mental, and gynecological health problems [7,24–27]. Also, it remains a symptom of gendered power relations [28,29], which may be a predictor of women's health [30,31], including stressful conditions [28,32], and nutritional status such as underweight, overweight, and obesity [18].

The issue of obesity is becoming a worldwide problem [33], increasingly also in developing countries [34]. Globally, overweight and obesity among female adults have increased from 29.8% to 38.0% between 1980 and 2013 respectively [32]. In Sub-Saharan Africa, the prevalence of overweight and obesity has been rising at an alarming rate [35], and women are the most affected [35]. In Zimbabwe, for instance, a recent study showed an increase in the prevalence of overweight and obesity from 25.0% in 2005 to 36.6% in 2015 [36]. The authors also observed socioeconomic and demographic differences in overweight and obesity among women of reproductive age. Differences in experiencing obesity and overweight among socioeconomic subgroups [37] may be linked to IPV in complex ways. For example, prior evidence suggests that abused women may end up suffering from depression [38], and may hence seek consolation in overeating [39]. In rich food environments, they tend to consume energy-dense foods [40], which has been shown to be a risk factor for obesity [18,40]. Furthermore, there is evidence that physical and sexual violence against women may predict excessive weight gain and poor nutrition [41,42], where some abused women tend to suffer from depression, increased anxiety, loss of appetite, and eating disorders with limited caloric intake [43,44]. The stress suffered by abused women has been shown to increase oxidative stress and metabolic syndrome including obesity and cardiovascular disease [44,45], which are also risk factors for anemia and underweight [30,38]. IPV thus contributes to the risk of poor nutrition outcomes, especially where abusive male partners withhold food as a form of punishment to their female partners [46].

From the above discussions, it is clear from the literature that there is a relationship between IPV and women's health [47,48]. While some studies have examined the relationship between dietary knowledge, the attitude of behaviours, socio-demographic factors, and IPV [18,31,49,50], no study has investigated the association between IPV and the nutritional status of women in Zimbabwe. This study, therefore, sought to explore the relationship between IPV and nutritional status (i.e., underweight, overweight, and obesity) among women of reproductive age in Zimbabwe.

## Materials and methods

### Data

The analysis was based on pooled data of married/cohabiting women from the 2005/2006, 2010/2011, and 2015 Zimbabwe Demographic Health Surveys. The surveys were conducted by the Zimbabwe National Statistical Agency in collaboration with other international organizations, and they were nationally representative surveys of men and women in their reproductive age. The surveys employed a two-stage stratified cluster sampling technique based on census enumeration areas (EAs) and household samples in both rural and urban areas. The first stage was the selection of EAs with probability proportional to the size and the second stage involved household sampling. The analysis was limited to non-pregnant women of reproductive age with valid weight and height measurements. Pregnant women were excluded to avoid a misleading picture of the issue of overweight and obesity during pregnancy [36]. The samples after the exclusion were (survey year: 2005/2006; n = 4,031), (survey year: 2010/2011; n = 4,211) and (survey year: 2015; n = 4,766), with a pooled total (N = 13,008) for the final analysis.

### Measurement of the outcome variable

The outcome variable for this study was the nutritional status of women (i.e., underweight, normal weight, overweight, and obesity). The body mass index (BMI; weight (kg)/height (m) squared) was used to assess the nutritional status of women [51], and it is commonly used to classify underweight, overweight, and obesity in adults [52,53]. Respondents were classified according to the BMI criteria of the World Health Organisation (WHO): a) underweight, BMI < 18.5 kg/m$^2$; b) normal weight, BMI of 18.5–24.9 kg/m$^2$; c) overweight, BMI of 25.0–29.9 kg/m$^2$ and d) obesity, BMI $\geq$ 30.0 kg/m$^2$ [54]. In the surveys, participants' standing heights were measured using a measuring board and their weights were taken using the United Nations Children's Fund (UNICEF) electronic scale with a digital display.

### Independent variable

The independent variable in this study was IPV. The measurement of IPV in the surveys was based on the modified Conflict Tactics (CTS2) [23,55,56] and was administered following standard guidelines for research on domestic violence set by the World Health Organisation [57]. The questions posed to women measure included "did your husband/partner ever: slap, push, shake, punch, beat, kick or try to strangle you, throw something at you, threaten you using a harmful object?" These questions were used to derive physical violence. Sexual violence was assessed by the questions "did your husband/partner ever: physically force you to have sexual intercourse even when you did not want? Or force you with threats to perform any sexual acts you did not want?" Psychological violence was assessed using questions such as "did your husband/partner humiliate you in front of others, threaten to hurt you or those close to you with harm?" Responses were categorized as physical, emotional, sexual, physical or emotional,

physical or sexual, emotional or sexual, and physical, sexual or emotional. Answers in the affirmative were coded as "1", while women who never experienced any of the aforementioned forms of IPV were coded as "0".

## Covariates

In the adjusted regression models, we controlled for the following socio-demographic and economic variables: age (15–29, 20–24, 25–29, 30–34, 35–39, 40+); marital status (married, cohabiting); place of residency (rural, urban), educational level (no education, primary, secondary and higher); parity (<2, 2–3, 4–5, 6+); employment status (not currently employed, currently employed); and wealth index (poorest, poorer, middle, richer), guided by a directed acyclic graph (Fig 1). The wealth index in the DHS is usually computed using durable goods, household characteristics and basic services. All the variables were obtained from either the individual women's or the household questionnaires.

## Statistical analysis

First, basic descriptive statistics were performed to obtain the mean, frequency, and percentages of the dependent, independent, and some control variables. Second, percentages (%) were used to describe the prevalence of BMI $\geq 25.0$ kg/m$^2$ (overweight and obesity) and the various forms of IPV. Differences in prevalence were examined using chi-square test. Third, we estimated the prevalence of IPV among women who experienced at least one type of abuse (i.e. physical, sexual or emotional) by nutritional status (i.e., underweight, normal weight, overweight and obese). In the second part of the analysis, multinomial logistic regression models were used to examine the associations between the various forms of IPV and the nutritional status of women. The prevalence and adjusted odd ratios (aOR) with 95% confidence intervals (95% CI) was calculated using Stata Version 14 (Stata Corp, College Station, Texas, USA). The dataset was weighted to account for differences in the sampling design.

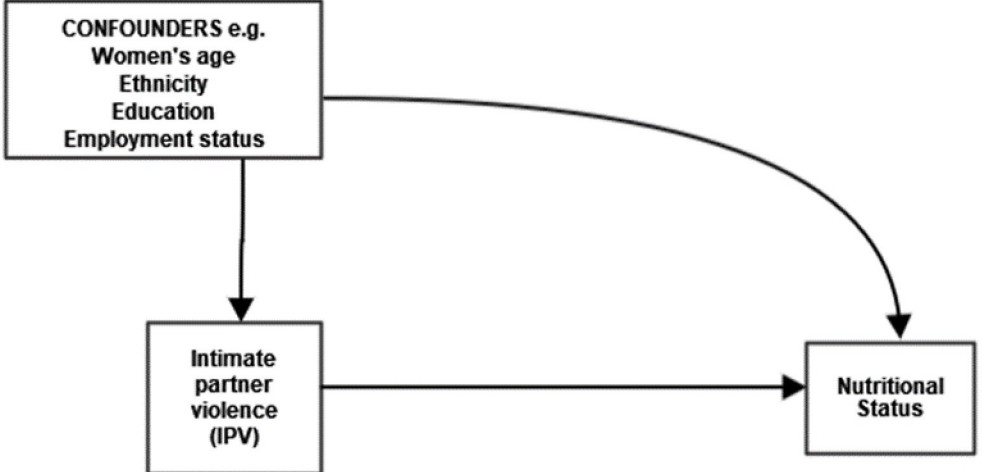

**Fig 1. Directed acyclic graph for the current study.**

## Results

### Distribution of selected characteristics

The distribution of respondents' characteristics is shown in Table 1. Overall, the mean age of women was approximately 30 years. Most women (64%) reported having secondary or higher education. On average, women had three live births, and about 67% lived in rural areas. Regarding economic status, more than half (61%) were not in paid employment, and 41% reported middle economic class.

The mean BMI of women was 24.3 kg/m$^2$ (Table 1). A high proportion of women had normal weight (59%), more than one-fifth were overweight (24%) and about 12% were obese. The results further showed that more than one-third (43%) of women reported to have ever experienced at least one form of intimate partner violence, and large proportions ever experienced physical (28%), emotional (28%), and sexual (13%) violence. More than one-third reported any physical or emotional violence (40%) and any emotional or sexual violence (33%).

In Table 2, the results of the prevalence of BMI $\geq$ 25.0 kg/m$^2$ (overweight and obesity) by intimate partner violence type are shown. In general, more than one-third (35%) of women who reported to have ever experienced at least one form of intimate partner violence (i.e., physical emotional, or sexual) had a BMI $\geq$ 25.0 kg/m$^2$ (p< 0.01). Similarly, more than one-third of women who ever experienced sexual (33%), any physical or emotional (34%), and any physical or sexual (33%) violence reported being overweight or obese. The overall proportion (%) of any form of intimate partner violence (i.e., physical, sexual, or emotional) was generally high (60%) among women who had normal weight (Fig 2). Meanwhile, the trend analysis by survey year showed a decline from 65.6% in 2005/2006 to 53.7% (Fig 3).

### Multinomial logistic regression

The adjusted odd ratios (aOR) and 95% confidence intervals for the associations between intimate partner violence and the nutritional status of women are shown in Table 3. The multinomial regression model estimated the relative risk ratios of the relationships between intimate partner violence and body mass index (BMI) comparing underweight, overweight, and obesity to normal weight. In the model, we adjusted for socioeconomic factors (categorical, as shown in Table 1) and other behavioural risk factors including smoking status (yes or no), alcohol consumption (yes or no), and media exposure (yes or no).

Results from Table 3 showed that women's exposure to any form of intimate partner violence was not significantly associated with the likelihood of being underweight or overweight relative to normal weight. However, women who had ever experienced at least one form of IPV (i.e., physical, emotional, or sexual) were more likely to be obese (aOR = 2.59; 95% CI = 1.05–6.39) relative to normal-weight women. Similarly, we found that women who had ever experienced all three forms of IPV more likely to be obese (aOR = 2.83; 95% CI = 1.28–6.25) relative to normal-weight women. The odds of being obese were also found to be higher among women with any prior exposure to emotional violence (aOR = 2.22; 95% CI = 1.16–4.13). Interestingly, the adjusted odds of being obese were lower among women who had ever experienced any emotional or sexual violence (aOR = 0.37; 95% CI = 0.18–0.73).

## Discussion

This is the first study to explore the association between Zimbabwean women's exposure to IPV and nutritional status using ZDHS data collected from 2005–2015. Although prior studies in Zimbabwe have examined trends in the prevalence of overweight and obesity [36] as well as associations between demographic characteristics, socioeconomic status, and IPV against

**Table 1. Percentage distribution of the characteristics of women (15–49 years) in Zimbabwe, pooled data, 2005–2015 (n = 13,008).**

| Variables | N = 13008 | % (95% CI) | Mean | (SD) | Min | Max |
|---|---|---|---|---|---|---|
| **Anthropometry** | | | | | | |
| BMI (Kg/m2) | | | 24.31 | (4.64) | 13.27 | 57.74 |
| Underweight, or BMI<18.5 | 665 | 5.11 (4.73–5.50) | | | | |
| Normal weight, or BMI 18.5≤BMI<25 | 7733 | 59.45 (58.59–60.29) | | | | |
| Over weight, or 25≤BMI<30 | 3068 | 23.59 (22.85–24.32) | | | | |
| Obese, or BMI≥30 | 1542 | 11.85 (11.3–12.4) | | | | |
| **Intimate Partner Violence, by type** | | | | | | |
| Physical | | | | | | |
| Ever | 3666 | 28.18 (27.41–28.96) | | | | |
| Never | 9342 | 71.82 (71.03–72.58) | | | | |
| Emotional | | | | | | |
| Ever | 3650 | 28.06 (27.28–28.84) | | | | |
| Never | 9358 | 71.94 (71.11–72.71) | | | | |
| Sexual | | | | | | |
| Ever | 1639 | 12.60 (12.03–13.18) | | | | |
| Never | 11369 | 87.40 (86.81–87.96) | | | | |
| Physical or Emotional | | | | | | |
| Ever | 5219 | 40.12 (39.2–40.96) | | | | |
| Never | 7789 | 59.88 (59.03–60.72) | | | | |
| Physical and Emotional | | | | | | |
| Ever | 2097 | 16.12 (15.49–16.76) | | | | |
| Never | 10911 | 83.88 (83.22–84.50) | | | | |
| Physical or Sexual | | | | | | |
| Ever | 4330 | 33.29 (32.47–34.10) | | | | |
| Never | 8678 | 66.71 (65.89–67.52) | | | | |
| Physical and Sexual | | | | | | |
| Ever | 957 | 7.50 (7.04–7.96) | | | | |
| Never | 12033 | 92.50 (92.03–92.95) | | | | |
| Emotional or Sexual | | | | | | |
| Ever | 4296 | 33.03 (32.21–33.84) | | | | |
| Never | 8714 | 66.97 (66.15–67.78) | | | | |
| Emotional and Sexual | | | | | | |
| Ever | 993 | 7.63 (7.18–8.10) | | | | |
| Never | 12015 | 92.37 (91.89–92.81) | | | | |
| Physical or Emotional or Sexual | | | | | | |
| Ever | 5615 | 43.17 (42.31–44.02) | | | | |
| Never | 7393 | 56.83 (55.97–57.68) | | | | |
| All three | | | | | | |
| Ever | 725 | 5.57 (5.18–5.98) | | | | |
| Never | 12283 | 94.43 (94.01–94.81) | | | | |
| **Sociodemographic controls** | | | | | | |
| Age | | | 30.36 | (7.96) | 15.0 | 49.0 |
| 15–19 | 824 | 6.33 (5.92–6.76) | | | | |
| 20–24 | 2696 | 20.73 (20.03–21.43) | | | | |
| 25–29 | 3038 | 23.35 (22.62–24.09) | | | | |
| 30–34 | 2639 | 20.29 (19.59–20.98) | | | | |
| 35–39 | 1776 | 13.65 (13.06–14.25) | | | | |

(*Continued*)

**Table 1.** (Continued)

| Variables | N = 13008 | % (95% CI) | Mean | (SD) | Min | Max |
|---|---|---|---|---|---|---|
| 40+ | 2035 | 15.64 (15.02–16.28) | | | | |
| Marital Status | | | | | | |
| Married | 12442 | 95.65 (95.28–95.99) | | | | |
| Cohabiting | 566 | 4.35 (4.01–4.71) | | | | |
| Parity | | | 2.80 | (1.87) | 0.0 | 13.0 |
| <2 | 3341 | 25.68 (24.93–26.44) | | | | |
| 2–3 | 5880 | 45.20 (44.34–46.06) | | | | |
| 4–5 | 2670 | 20.53 (19.83–21.23) | | | | |
| 6+ | 1117 | 8.59 (8.11–9.08) | | | | |
| Place of residence | | | | | | |
| Urban | 4340 | 33.36 (32.55–34.18) | | | | |
| Rural | 8668 | 66.64 (65.81–67.44) | | | | |
| Educational Level | | | | | | |
| No education | 363 | 2.79 (2.51–3.08) | | | | |
| Primary | 4330 | 33.29 (32.47–34.10) | | | | |
| Secondary and higher | 8315 | 63.92 (63.09–64.74) | | | | |
| Employment Status | | | | | | |
| Not currently employed | 7950 | 61.12 (60.27–61.95) | | | | |
| Currently employed | 5058 | 38.88 (38.04–39.72) | | | | |
| Wealth (Index) | | | | | | |
| Poorest | 2701 | 20.76 (20.01–21.47) | | | | |
| Poorer | 2482 | 19.08 (18.40–19.76) | | | | |
| Middle | 5394 | 41.47 (40.61–42.31) | | | | |
| Richer | 2431 | 18.69 (18.02–19.36) | | | | |

women [1], no study has investigated the complex relationship between IPV and nutritional status (i.e., underweight, overweight, and obesity) of women in the country. Moreover, the prevalence of both IPV and overweight is high in Zimbabwe [1,36,58,59], which makes the country an appropriate setting for this study.

Overall, the findings revealed that more than one-third (43%) of women reported to have ever experienced at least one form of intimate partner violence, which is higher than the global estimated prevalence of 30% [1,60]. These findings are consistent with previous studies in Zimbabwe [1,23,58,61,62] and other Sub-Saharan African countries [63,64]. Some of the risks for the high and increasing prevalence of IPV in developing countries have been attributed to cohabitation [65], rural residence [66,67], and low economic status [68–70]. Poverty on the other hand has been shown to be a determinant of IPV [71,72] as poor women tend to heavily depend on their partners [69,72,73], which may limit their bargaining powers.

Regarding the various forms of IPV, we found emotional and sexual violence to be the most popular forms of violence against women [58,62]. Sexual violence may be low due to underreporting of such abuses in Africa [67,74], stemming from traditional norms and beliefs [75].

The findings further revealed that women of reproductive age are at high risk of excess weight [35,76,77], as more than one-fifth reported being overweight and about 12% obese. Several studies have reported overweight and obesity to be on the rise in developing countries [33,35,36], and risk factors such as high economic status, urban residence [78,79], and, indeed, intimate partner violence [80,81] have been implicated.

**Table 2. Prevalence of BMI $\geq$ 25.0 kg/m$^2$ (overweight and obesity) among women of reproductive age (15–49 years) by intimate partner violence type, Zimbabwe, pooled data, 2005–2015.**

| Variables | BMI$\geq$25 Kg/m2 (%) | P value* |
|---|---|---|
| **Intimate Partner Violence, by type** | | |
| Physical | | < 0.001 |
|   Ever | 31.86 (30.37–33.38) | |
|   Never | 36.84 (35.87–37.82) | |
| Emotional | | 0.244 |
|   Ever | 34.66 (33.12–36.21) | |
|   Never | 35.75 (34.77–36.72) | |
| Sexual | | < 0.05 |
|   Ever | 33.07 (30.83–35.38) | |
|   Never | 35.78 (34.90–36.66) | |
| Physical or Emotional | | < 0.001 |
|   Ever | 33.89 (32.62–35.19) | |
|   Never | 36.62 (35.41–37.55) | |
| Physical and Emotional | | < 0.001 |
|   Ever | 31.66 (29.70–33.68) | |
|   Never | 36.16 (35.26–37.07) | |
| Physical or Sexual | | < 0.001 |
|   Ever | 32.49 (31.11–33.90) | |
|   Never | 36.91 (35.89–37.93) | |
| Physical and Sexual | | < 0.05 |
|   Ever | 31.17 (28.24–34.05) | |
|   Never | 35.79 (34.94–36.65) | |
| Emotional or Sexual | | < 0.05 |
|   Ever | 34.08 (32.67–35.50) | |
|   Never | 36.11 (35.10–37.12) | |
| Emotional and Sexual | | 0.538 |
|   Ever | 34.54 (31.64–37.55) | |
|   Never | 35.51 (34.66–36.37) | |
| Physical or Emotional or Sexual | | < 0.01 |
|   Ever | 35.44 (32.66–37.10) | |
|   Never | 36.62 (35.52–37.72) | |
| All three | | 0.130 |
|   Ever | 32.82 (29.50–36.33) | |
|   Never | 35.55 (34.75–36.44) | |

Note -

* p values are based on the χ2 test, data are % (95% CI—Clopper-Pearson).

Both intimate partner violence against women and obesity are growing health problems in low and middle-income countries (LMICs) [33–35,52,64,76,81]. Our findings showed that women who had ever experienced any form of IPV were more likely to be obese. Prior research have linked stressors including IPV with obesity [82]. It has been shown that stressful conditions may lead to the development of obesity through several mechanisms and pathways including increased hormone release [83,84], which can increase food cravings. [85,86]. In a study, Torres and Nowson (2007) found increased rate of obesity among people who face mild stressors [18]. This may be due to overeating and consumption of food that are in high calories

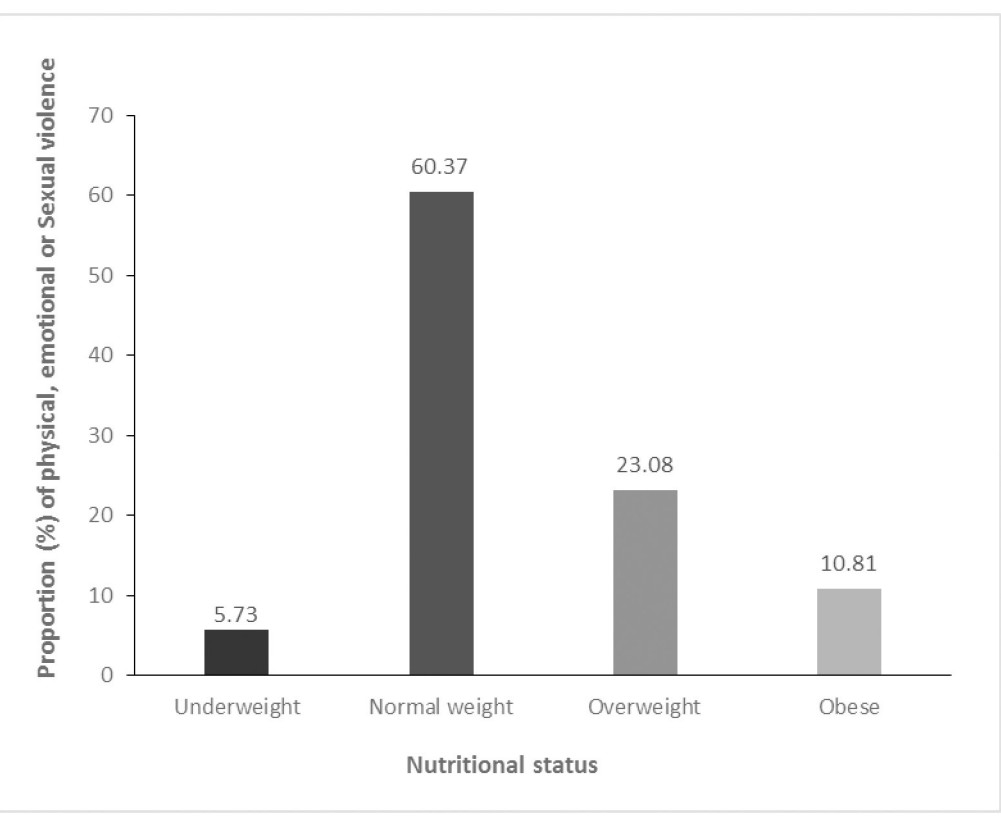

**Fig 2. Proportion (%) of physical, emotional or sexual violence against women of reproductive age (15–49 years) by nutritional status, Zimbabwe, pooled data, 2005–2015.**

or sugar [87,88], which may affect behavioural patterns such as sleep and physical activity [89]. There is some evidence that obesity affects women's participation in daily routines [90–92] which can affect their participation in the labour market [86], and also impact other health outcomes [85,93].

Surprisingly, we did not find any significant association between IPV and underweight, relative to normal weight. While this finding is consistent with some studies [77,94], others suggest that exposure to IPV increases the odds of being underweight [94,95]. These inconsistent findings may be attributed to study population, demographic and socioeconomic contexts [18,30,94]. Meanwhile, the positive association between IPV and underweight has been associated with dietary behaviours characterized by substance abuse, insufficient calorie intake, or reduced food intake [30]. Furthermore, abusive partners may withhold food from victims, as a form of punishment that can negatively affect their weight [18,30]. These inconsistent findings call for future research to explore this issue closely.

IPV and poor nutrition (underweight and overweight) are major determinants of health [96,97], especially among women of reproductive age [98,99]. While obesity is a risk factor for non-communicable diseases such as diabetes and hypertension [100–102], IPV has been linked with mental health problems including traumatic stress [15,103,104] and injury [5,24,105]. These findings, including the results presented in the current study, should be taken into account for the development of policies aiming for the promotion of peace and security of women. Such policies need to address gender-related health issues as well as

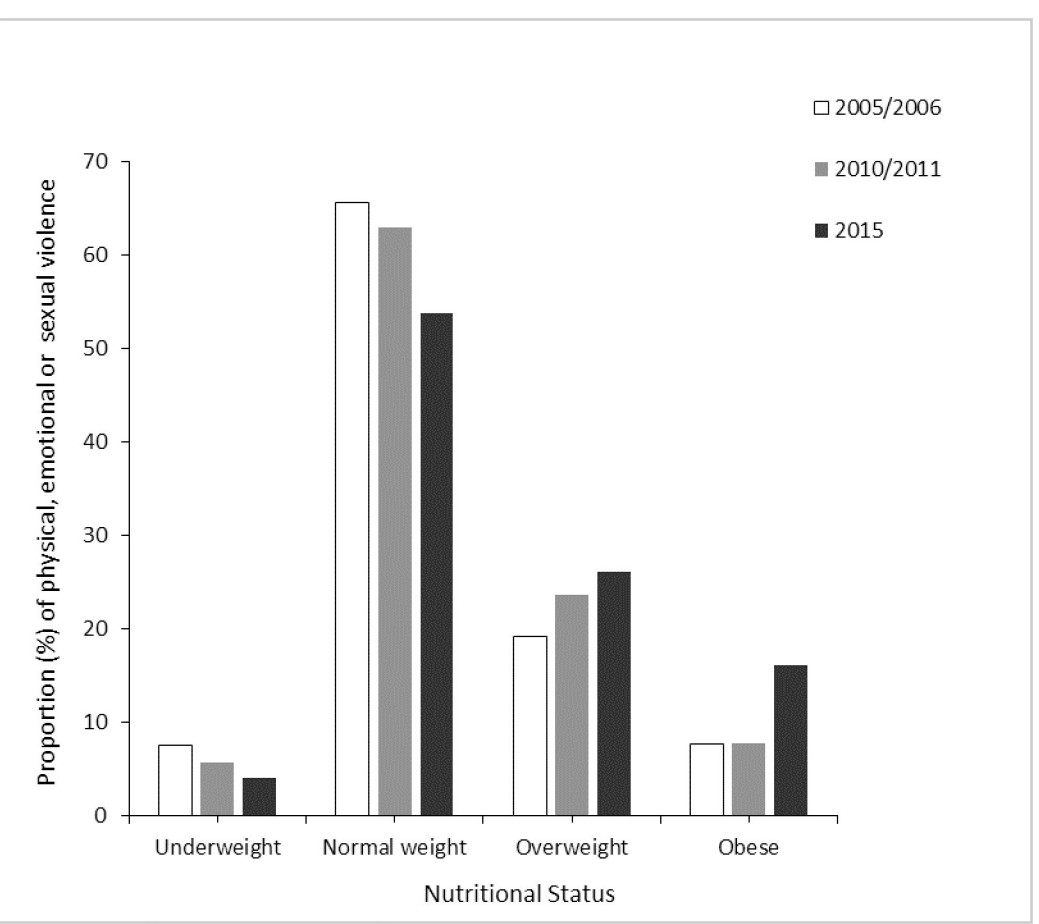

**Fig 3. Proportion (%) of physical, emotional or sexual violence against women of reproductive age (15–49 years) by nutritional status and survey year, Zimbabwe.**

opportunities and pathways to reduce gender inequity and gendered social and health problems including IPV.

## Strengths and limitations

The major strength of this study was that a nationally representative sample was used, where participants were sampled using probability sampling methods [23]. The range of relevant questions in the survey allowed for a detailed assessment of the IPV-obesity link in a large sample of women from Zimbabwe. Nonetheless, there are some limitations. First, due to the cross-sectional design of the DHS data, causality of associations between variables cannot be established. Longitudinal studies on exposure to IPV and the association with adverse health outcomes would be better suited for causal interpretation, although the currently available survey data already provide some convincing insights into the problem under investigation. Second, it has been shown that exposure to violence during childhood may increase subsequent exposures in adulthood [80,106,107], which may lead to excess weight. However, the study lacks data on violence experienced during childhood. Third, this study used secondary data, hence, information on other imperative behavioural factors such as nutritional history and physical inactivity that might have explained the prevalence of excess weight in the study sample was

**Table 3. Multinomial logistic regression of the association between intimate partner violence and nutritional status of women (15–49 years), Zimbabwe, pooled data, 2005–2015.**

| Variables | Underweight—RRR (95%) | Overweight—RRR (95%) | Obese—RRR (95%) |
|---|---|---|---|
| **Intimate Partner Violence, by type** | | | |
| Physical | | | |
| Never (ref) | 1 | 1 | 1 |
| Ever | 1.31 (0.66–2.62) | 0.93 (0.65–1.32) | 0.97 (0.59–1.59) |
| Emotional | | | |
| Never (ref) | 1 | 1 | 1 |
| Ever | 1.41 (0.61–2.14) | 1.32 (0.91–1.91) | 2.22 (1.16–4.13)** |
| Sexual | | | |
| Never (ref) | 1 | 1 | 1 |
| Ever | 1.04 (0.71–1.53) | 1.14 (0.91–1.44) | 1.29 (0.92–1.81) |
| Physical or Emotional | | | |
| Never (ref) | 1 | 1 | 1 |
| Ever | 0.67 (0.25–1.78) | 0.83 (0.48–1.42) | 0.51 (0.22–1.81) |
| Physical and Emotional | | | |
| Never (ref) | 1 | 1 | 1 |
| Ever | 1.13 (0.88–1.45) | 0.93 (0.80–1.07) | 0.76 (0.61–0.94) |
| Physical or Sexual | | | |
| Never (ref) | 1 | 1 | 1 |
| Ever | 0.81 (0.37–1.76) | 0.92 (0.61–1.37) | 0.71 (0.40–1.25) |
| Physical and Sexual | | | |
| Never (ref) | 1 | 1 | 1 |
| Ever | 1.03 (0.59–1.80) | 0.80 (0.58–1.12) | 0.44 (0.25–0.79) |
| Emotional or Sexual | | | |
| Never (ref) | 1 | 1 | 1 |
| Ever | 0.78 (0.38–1.59) | 0.75 (0.49–1.14) | 0.37 (0.18–0.73)*** |
| Emotional and Sexual | | | |
| Never (ref) | 1 | 1 | 1 |
| Ever | 0.90 (0.48–1.67) | 1.14 (0.85–1.54) | 1.01 (0.67–1.53) |
| Physical or Emotional or Sexual | | | |
| Never (ref) | 1 | 1 | 1 |
| Ever | 1.87 (0.64–5.43) | 1.31 (0.72–2.37) | 2.59 (1.05–6.39)* |
| All three | | | |
| Never (ref) | 1 | 1 | 1 |
| Ever | 1.11 (0.44–2.82) | 1.23 (0.74–2.04) | 2.83 (1.28–6.25) |

Notes: aOR- adjusted Odd Ratio. Model adjusted for women's age, marital status, education, ethnicity, and parity, place of residence, employment status, wealth, smoking status, alcohol consumption, and media exposure.

not available. Fourth, DHS measures self-reported IPV, and this may under estimate IPV among participants in our sample. Finally, it is likely that IPV reporting is hampered by issues of privacy, shame, etc. This can lead to information bias, hence additional approaches to validate and enhance information on IPV experiences need to be considered [108–110].

## Conclusion

The study findings show that women of reproductive age in Zimbabwe are at high risk of both IPV and excess weight. Moreover, we found a positive relationship between exposure to at

least one form of IPV and obesity. Public health interventions that target the well-being, empowerment and development of women are needed to address the complex issue of IPV and adverse health outcomes, including obesity. Legal, social and health institutions should collaborate to develop and implement appropriate intervention measures.

## Author Contributions

**Conceptualization:** Jeanette Iman'ishimwe Mukamana, Nicholas Kofi Adjei.

**Data curation:** Jeanette Iman'ishimwe Mukamana, Nicholas Kofi Adjei.

**Formal analysis:** Jeanette Iman'ishimwe Mukamana, Nicholas Kofi Adjei.

**Methodology:** Nicholas Kofi Adjei.

**Supervision:** Pamela Machakanja, Hajo Zeeb, Nicholas Kofi Adjei.

**Writing – original draft:** Jeanette Iman'ishimwe Mukamana.

**Writing – review & editing:** Jeanette Iman'ishimwe Mukamana, Pamela Machakanja, Hajo Zeeb, Sanni Yaya, Nicholas Kofi Adjei.

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
