## [Decision Letter · Decision Letter 0]

11 Jan 2022

PONE-D-21-11105

Investigating the associations between intimate partner violence and nutritional status of women in Zimbabwe

PLOS ONE

Dear Dr. Adjei,

Thank you for submitting your manuscript to PLOS ONE. After careful consideration, we feel that it has merit but does not fully meet PLOS ONE’s publication criteria as it currently stands. Therefore, we invite you to submit a revised version of the manuscript that addresses the points raised during the review process.

The manuscript has been evaluated by two reviewers, and their comments are available below.

The reviewers have raised a number of concerns that need attention. They feel the introduction should outline the state of the art regarding studies on IPV and nutritional status. The reviewers also request improvements to the reporting of methodological aspects of the study, for example, regarding the use of the full Revised Conflict Tactics (CTs) instrument for the IPV measurement.

Could you please revise the manuscript to carefully address the concerns raised?

We look forward to receiving your revised manuscript.

Kind regards,

Lorena Verduci

Staff Editor

PLOS ONE

Journal Requirements:

2. Thank you for stating the following financial disclosure: "No"

3. Thank you for stating the following in your Competing Interests section: "No"

Reviewers' comments:

Reviewer's Responses to Questions

**Comments to the Author**

1. Is the manuscript technically sound, and do the data support the conclusions?

Reviewer #1: Partly

Reviewer #2: Yes

2. Has the statistical analysis been performed appropriately and rigorously? 

Reviewer #1: Yes

Reviewer #2: Yes

3. Have the authors made all data underlying the findings in their manuscript fully available?

Reviewer #1: No

Reviewer #2: No

4. Is the manuscript presented in an intelligible fashion and written in standard English?

Reviewer #1: Yes

Reviewer #2: Yes

5. Review Comments to the Author

Reviewer #1: The thematic of the manuscript is interesting and relevant, given the scarcity of studies on the relationship between IPV and inadequate nutritional status.

However, for publication the paper needs some improvements.

ABSTRACT

I suggest revising the wording of the passage “Relative to normal weight, women who had ever experienced at least one form of IPV (i.e., physical, emotional, or sexual) were more likely to be obese (aOR = 2.59; 95% CI = 1.05–6.39). Women’s exposure to any form of intimate partner violence was not significantly associated with the likelihood of being underweight or overweight relative to normal weight.”, because it is confusing since in the conclusion of the abstract, you state that “we found a positive relationship between IPV and obesity.”

INTRODUCTION

The introduction lacks information about the state of the art regarding studies on intimate partner violence and nutritional status in order to point out what has been studied and the gaps.

METHODS

The authors could make it clear whether the full Revised Conflict Tactics (CTs) instrument or single questions were used for the IPV measurement. If the authors used single questions, this should be included as one of the limitations of the paper and how this form of data collection impacts the findings.

Is the marital status only these two answer options (married, cohabiting)? And women who are dating and the partner does not reside at the household

RESULT

The main finding of the paper which is the positive association of emotional IPV with obesity is "erased" by the other analyses which involve the possibility of having 1 of the two types of violence (physical OR emotional; physical OR sexual, ...). I think the analysis could be more objective and contemplate the role of each type of violence separately or their coexistence.

Table 1: insert 95% confidence interval

Table 2: insert 95% confidence interval

In table 2 it would be more interesting to bring the specific types of violence (emotional, physical and sexual) and the co-occurrence emotional AND physical, emotional AND sexual and this way on

Table 3

it would be more interesting to bring the specific types of violence (emotional, physical and sexual) and the co-occurrence emotional AND physical, emotional AND sexual and this way on

Figure 1

It would be interesting to bring the proportion of physical, emotional or sexual violence against women of reproductive age (15–49 years) by nutritional status by year 2005/2006,

108 2010/2011, and 2015

DICUSSION

The discussion could delve into the possible mechanisms that may explain the relationship found (IPV positively associated with obesity).

Insert as a limitation how to measure the exposure of interest (IPV).

Reviewer #2: Introduction

The introduction is short and I think this is a good option. However, I suggest contextualizing better the IPV situation in Zimbabwe.

Method

The covariates measurement could be described in the method

A theoretical model (DAG) could be shown in the methods section

Table 1 – Table 1 could inform the number of women in each category, and the confidence interval.

Table 2 – table 2 could present the prevalence by BMI status

6. PLOS authors have the option to publish the peer review history of their article (what does this mean?). If published, this will include your full peer review and any attached files.

Reviewer #1: No

Reviewer #2: **Yes: **Tatiana Henriques Leite

---

## [Author Response · Author response to Decision Letter 0]

3 Apr 2022

Reviewer #1: Overall comments

The thematic of the manuscript is interesting and relevant, given the scarcity of studies on the relationship between IPV and inadequate nutritional status.

Response: Thank you. 

1.1 Major Comments: Abstract

1.1a. I suggest revising the wording of the passage “Relative to normal weight, women who had ever experienced at least one form of IPV (i.e., physical, emotional, or sexual) were more likely to be obese (aOR = 2.59; 95% CI = 1.05–6.39). Women’s exposure to any form of intimate partner violence was not significantly associated with the likelihood of being underweight or overweight relative to normal weight.”, because it is confusing since, in the conclusion of the abstract, you state that “we found a positive relationship between IPV and obesity.”

Response: Thank you for this comment, we have now revised the wording in the abstract.

Page 1, line39-43: “Relative to normal weight, women who had ever experienced at least one form of IPV (i.e., physical, emotional, or sexual) were more likely to be obese (aOR = 2.59; 95% CI = 1.05–6.39). Women’s exposure to any form of intimate partner violence was not significantly associated with the likelihood of being underweight or overweight relative to normal weight.” And Line 45-46: “between exposure to at least one form IPV and obesity.”

1.2. Comment: Introduction

1.2a.The introduction lacks information about the state of the art regarding studies on intimate partner violence and nutritional status in order to point out what has been studied and the gaps.

Response: Thank you. We have added more literature to point out what has been studied and the existing gaps regarding research on intimate partner violence and nutritional status.

Page 2, line 68-73: “The prevalence of IPV is high in developing countries [1]. However, there is evidence of cross-country variations [20,21], where Zimbabwe has been found to be one of the countries in sub-Saharan Africa with the highest prevalence of IPV [21,22]. It is estimated that approximately 35% of women had experienced physical violence from the age of 15 and 14% had experienced sexual violence [23]. In a recent study in Zimbabwe, Mukamana and colleagues found a substantial rise in the prevalence of IPV from 40.9% in 2010 to 43.1% in 2015 [1].”

Line 83-86: “In Zimbabwe, for instance, a recent study showed an increase in the prevalence of overweight and obesity from 25.0% in 2005 to 36.6% in 2015 [36]. The authors also observed socioeconomic and demographic differences in overweight and obesity among women of reproductive age.”

1.3. Comment: Methods

1.3ai.The authors could make it clear whether the full Revised Conflict Tactics (CTs) instrument or single questions were used for the IPV measurement. If the authors used single questions, this should be included as one of the limitations of the paper and how this form of data collection impacts the findings.

Response: Thank you, the Modified Conflict Tactic scale was used for IPV measurement, in line with previous studies. 

Page 3, line 132-135: “We used the measurement of IPV in the surveys that was based on the modified Conflict Tactics (CTS2).”

1.3aii.Is the marital status only these two answer options (married, cohabiting)? And women who are dating and the partner does not reside at the household. 

Response: Thanks. This study considerered only those in unions (living together with their partners) – i.e., either married or cohabiting.

1.4. Comments: Results

1.4ai. RESULT

The main finding of the paper which is the positive association of emotional IPV with obesity is "erased" by the other analyses which involve the possibility of having 1 of the two types of violence (physical OR emotional; physical OR sexual, ...). I think the analysis could be more objective and contemplate the role of each type of violence separately or their coexistence.

Response: Thank you for your comment. We have now assessed the co-occurrence of the various forms of violence. (see table 1, 2 and 3) and the result section. 

Page 5-6, line 217-219: “Similarly, we found that women who had ever experienced all three forms of IPV more likely to be obese (aOR = 2.83; 95% CI = 1.28–6.25) relative to normal-weight women.”

1.4aii. Table 1: insert 95% confidence interval

Response: We have now added 95% confidence interval (See table1) 

1.4aiii.Table 2: insert 95% confidence interval

Response: Thanks. We have now added 95% confidence interval (See table1)

1.4aiv. In table 2 it would be more interesting to bring the specific types of violence (emotional, physical, and sexual) and the co-occurrence emotional AND physical, emotional AND sexual and this way on. 

Response: Thank you for your comment. We have now included the co-occurrences of the various forms of violence (table 2) 

1.4av.Table 3. It would be more interesting to bring the specific types of violence (emotional, physical, and sexual) and the co-occurrence emotional AND physical, emotional AND sexual and this way on

Response: Thanks. We have now included the co-occurrences of the various forms of violence (table 3)

1.4avi. Figure 1

It would be interesting to bring the proportion of physical, emotional, or sexual violence against women of reproductive age (15–49 years) by nutritional status the year 2005/2006, 2010/2011, and 2015

Response: Thank you. We have now included the proportion of physical, emotional, or sexual violence against women of reproductive age (15–49 years) by nutritional status the year 2005/2006, 2010/2011, and 2015 ( See figure 2). 

1.5 Comments: Discussion

1.5ai.The discussion could delve into the possible mechanisms that may explain the relationship found (IPV positively associated with obesity)

.

Response: Thanks for this comment: we have now added more possible mechanisms to explain the positive association of IPV with obesity.

Page 6, line 252-259: “Prior research have linked stressors including IPV with obesity [82]. It has been shown that stressful conditions may lead to the development of obesity through several mechanisms and pathways including increased hormone release [83,84], which can increase food cravings. [85,86]. In a study, Torres and Nowson (2007) found increased rate of obesity among people who face mild stressors [18]. This may be due to overeating and consumption of food that are in high calories or sugar [87,88], which may affect behavioural patterns such as sleep and physical activity [89].” 

1.5aii. Insert as a limitation how to measure the exposure of interest (IPV).

Response: Thank you. We have included this suggestion in the limitation section.

 Page 7, line 293-294: “Fourth, DHS measures self-reported IPV, and this may under

 estimate IPV among participants in our sample”

Reviewer #2: Major Comments

1.1. Comment: Introduction

1.1a.The introduction is short and I think this is a good option. However, I suggest contextualizing better the IPV situation in Zimbabwe.

Response: Thank you for your comment. We have revised the introduction to contextualize the IPV situation in Zimbabwe and added more literature to contextualize this issue in Zimbabwe.

Page 2, line 68-73: “The prevalence of IPV is high in developing countries [1]. However, there is evidence of cross-country variations [20,21], where Zimbabwe has been found to be one of the countries in sub-Saharan Africa with the highest prevalence of IPV [21,22]. It is estimated that approximately 35% of women had experienced physical violence from the age of 15 and 14% had experienced sexual violence [23]. In a recent study in Zimbabwe, Mukamana and colleagues found a substantial rise in the prevalence of IPV from 40.9% in 2010 to 43.1% in 2015 [1].”

1.2 Comment: Method

1.2a.The covariates measurement could be described in the method.

Response: Thank you. We have now described the computed measures. 

Page 4, line 153-155. “The wealth index in the DHS is usually computed using durable goods, household characteristics and basic services”

1.2b.A theoretical model (DAG) could be shown in the methods section

Response: We have now included DAG in the method section (See figure 1)

1.2c.Table 1 – Table 1 could inform the number of women in each category, and the confidence interval.

Response: Thank you, we have included the number of women in each category, and the confidence interval (see table 1)

1.2d.Table 2 – table 2 could present the prevalence by BMI status

Response: Thank you for your comment. Table 2 is presented by BMI status (i.e., over weight and obesity vs the other groups). We feel that dichotomising it is easier for interpretation.

---

## [Decision Letter · Decision Letter 1]

13 Jul 2022

Investigating the associations between intimate partner violence and nutritional status of women in Zimbabwe

PONE-D-21-11105R1

Dear Dr. Adjei,

We’re pleased to inform you that your manuscript has been judged scientifically suitable for publication and will be formally accepted for publication once it meets all outstanding technical requirements.

Kind regards,

Carla Pegoraro

Division Editor

PLOS ONE

Additional Editor Comments (optional):

Reviewers' comments:

Reviewer's Responses to Questions

**Comments to the Author**

1. If the authors have adequately addressed your comments raised in a previous round of review and you feel that this manuscript is now acceptable for publication, you may indicate that here to bypass the “Comments to the Author” section, enter your conflict of interest statement in the “Confidential to Editor” section, and submit your "Accept" recommendation.

Reviewer #2: All comments have been addressed

2. Is the manuscript technically sound, and do the data support the conclusions?

Reviewer #2: Yes

3. Has the statistical analysis been performed appropriately and rigorously? 

Reviewer #2: Yes

4. Have the authors made all data underlying the findings in their manuscript fully available?

Reviewer #2: Yes

5. Is the manuscript presented in an intelligible fashion and written in standard English?

Reviewer #2: Yes

6. Review Comments to the Author

Reviewer #2: I have been giving a review before. The authors attend my request. However, the DAG is not exactly what I was expecting. So, I recommend two websites to the Authors. It is not a mandatory request. But the confounders are been addressed with this technique currently.

https://cran.r-project.org/web/packages/ggdag/vignettes/intro-to-dags.html

http://www.dagitty.net/dags.html

7. PLOS authors have the option to publish the peer review history of their article (what does this mean?). If published, this will include your full peer review and any attached files.

Reviewer #2: **Yes: **Tatiana Henriques Leite

---

## [Editor Report · Acceptance letter]

15 Jul 2022

PONE-D-21-11105R1 

Investigating the associations between intimate partner violence and nutritional status of women in Zimbabwe 

Dear Dr. Adjei:

I'm pleased to inform you that your manuscript has been deemed suitable for publication in PLOS ONE. Congratulations! Your manuscript is now with our production department. 

Kind regards, 

on behalf of

Dr Carla Pegoraro 

Staff Editor

PLOS ONE